# Enhancement of the Josephson Current in a Quantum Dot Connected to Majorana Nanowires

**DOI:** 10.3390/nano13091482

**Published:** 2023-04-26

**Authors:** Feng Chi, Qiang-Sheng Jia, Jia Liu, Qing-Guo Gao, Zi-Chuan Yi, Li-Ming Liu

**Affiliations:** 1School of Electronic and Information Engineering, UEST of China, Zhongshan Institute, Zhongshan 528400, Chinaqggao@zsc.du.cn (Q.-G.G.); yizichuan@zsc.edu.cn (Z.-C.Y.); liulmxps@zsc.edu.cn (L.-M.L.); 2School of Science, Inner Mongolia University of Science and Technology, Baotou 014010, China; jialiu@imust.edu.cn

**Keywords:** Josephson current, pure spin current, quantum dot, Majorana nanowires, spin-polarized coupling

## Abstract

We investigate the behavior of the Josephson current in a system composed of a quantum dot (QD) sandwiched between two nanowires by using the nonequilibrium Green’s function technique. We consider that the nanowires are in proximity to s-wave superconducror substrates, and Majorana bound states (MBSs) are induced at their ends. It is also assumed that the two nanowires are not aligned in the same orientation, but form a bent angle with respect to each other. It is found that when only one spin state on the QD is coupled to the left nanowire, the Josephson current is the typical sinusoidal function of the phase difference between the two nanowires. If both spin states hybridize to the MBSs with equal coupling strengths, the Josephson current then is not a sinusoidal function of the phase difference. In particular, when the bent angle between the two nanowires is π/2 and the two modes of the MBSs in each nanowire are decoupled from each other, the Josephson current is enhanced by about twenty times in magnitude as compared to the former case. Moreover, the simultaneously enhanced currents of the two spin directions are of the same magnitude but flow in opposite directions and they induce a large pure spin current. Our results also show that this abnormally enhanced Josephson current will be suppressed by a vertical magnetic field applied to the QD.

## 1. Introduction

Majorana bound states (MBSs) prepared at various platforms have received much attention during the last two decades due to their exotic properties, such as the topologically protected surface states [1,2]. These MBSs emerge at the ends of a nanowire (Majorana nanowire) as a pair of states at zero energy [3]. Two modes of MBSs can also be viewed as a pair of quasiparticles of equal-weight superpositions of electron and hole states. Therefore, MBSs are of their own anti-particles and electrically neutral [1,2]. Moreover, MBSs are anyons obeying non-Abelian statistics [2], enabling them to be exploited for fault-tolerant quantum information processing immune against the most common sources of decoherence [1,2,3]. Researchers have also put forward some other interesting applications of MBSs in addition to quantum computation, such as spintronics [4], optical devices [5] and thermoelectric instruments [6,7,8]. MBSs were theoretically predicted in 2008 to be prepared at the ends of nanowires fabricated from topological superconductors with Rashba-type spin–orbit coupling. This requires that the nanowire is in proximity of an s-wave superconductor in the presence of high-intensity magnetic fields [9]. Some other schemes have also proven that nonlocal MBSs are realizable in Josephson junctions [10,11,12,13,14].

The detection of MBSs is also a currently interesting challenge. One of the most reliable signatures of MBSs is the observation of the zero-energy state exhibited as a zero-bias abnormal peak in the electric differential conductance [15], whose peak value should reach its quantized maximum of $2e^{2}/h$. However, some other mechanisms, such as Andreev bound states (ABSs) due to disorder [16,17,18,19], quantum dots (QD) on nanowires [20,21,22,23] or the Kondo effect [24], may also induce a zero-bias conductance abnormality that is stable with respect to the experimental parameters. Further signatures expected for MBSs are phase-coherent transport of the quasiparticles, the 4π Josephson effect and the topological Kondo effect [24]. In time domain measurements, MBSs should furthermore show distinct behavior due to coherent MBS–MBS direct hybridization [5,25]. Some other work has demonstrated that the MBSs will change sign or exhibit a significant increase in the thermopower or thermoelectric efficiency in systems composed of QDs side-coupled to MBSs [6,7,8].

It has been demonstrated both theoretically and experimentally that MBSs can also be detected with the help of Josephson junctions formed by two topological superconductors. This is because they induce Andreev reflection processes characterized by the zero-bias abnormality in the conductance, as mentioned above [11,12,13,14,15]. In such structures, MBSs in a superconductor hybridize with each other and form ABSs [26,27]. It is well known that ABSs are states arising in semiconductors in close proximity, leading to a sharp configuration of the density of states within the energy gap in the superconductors [17,18,19,26,27,28,29,30,31,32]. There are various platforms to generate these subgap states, such as semiconductor QDs coupled to a superconductor. In such schemes, discrete energy levels of the QD couple to the superconductor and the ABSs exhibit various different properties depending on the size of the superconducting pairing, the coupling amplitude to the superconductor and the spin of electrons in the QDs, as well as the charging energy of the QDs [29,30,31,32]. As a result, some works have been devoted to the Josephson current carried by ABSs through a QD connected to nanowires which are in proximity to s-wave superconductors in order to generate MBSs [26,27]. The Josephson current is very different from cases where the QD is sandwiched between conventional superconductors, for example, its value is enhanced when the QDs level is not aligned to the Fermi energy [26,27].

In previous work concerning a QD connected to two Majorana nanowires, the MBSs were assumed to interact with spinless electrons or electrons with a certain spin direction [9,10,20]. This is reasonable because of the MBSs helical properties and was demonstrated in earlier theoretical and experimental work. On the other hand, recent works have shown that electrons with both spin directions may hybridize to the MBSs [21,22,23,26,27,33,34,35], which will exert a significant influence on current tunneling through the system via, for example, quantum interference between electrons of opposite spin directions. Experimentally, it was shown that [33] spin-dependent MBS–QD coupling strengths are related to the length of the spin–orbit coupling in the nanowires. Therefore, variations in the distance between the QD and the nanowires hosting MBSs will enable the adjustment of coupling amplitudes, leading to a fully spin-polarized coupling strength. In this manuscript, we study the possible influences of spin-dependent QD–MBS coupling on the Josephson current. The schematic plot is shown in Figure 1A. Specifically, the MBSs in the left nanowire couple to electrons in the QD with spin-resolved coupling strengths, which are experimentally determinable [23,33]. As for the right Majorana nanowire, it may be not straight with respect to the left one in experiments, and in this situation, the QD–MBS coupling strengths also depend on the electron spin direction. Our numerical results show that if only spin-up or spin-down electrons interact with the MBSs in the left nanowire, the Josephson current is a typical sinusoidal function of the phase difference, which is consistent with previous work. When the MBS is hybridized simultaneously to electrons of both spin directions with symmetrical coupling amplitudes however, the sinusoidal lineshape of the Josephson current varying with the phase difference is destroyed, accompanied by an abrupt enhancement in the current. Now, an equal number of electrons at different spin states flow in opposite directions and lead to a large pure spin current. We attribute this result to the perfect constructive quantum interference effect between electrons with opposite spin directions. We emphasize that this abnormally enhanced Josephson current occurs when the left and right nanowires are perpendicular to each other (bent angle θ=π/2). If only one of the Majorana nanowires couples to the QD with spin-dependent coupling due to the non-zero bent angle [26,27], there is actually only one spin direction current, which is quite different from the present case. Moreover, we find these enhanced spin-resolved Josephson currents are equal in amplitude but opposite in direction, and thus lead to a large pure spin current which has been discussed in some previous works [36,37,38]. Note that the mechanism of the spin current in the present manuscript is quite different from those in previously published ones.

## 2. Model and Method

We divide the system’s Hamiltonian into three parts, $H=H_{QD}+H_{MBSs}+H_{T}$ [20,21,26,27,39], in which $H_{QD}=\sum_{\sigma }\varepsilon_{\sigma }d_{\sigma }^{†}d_{\sigma }$ is the QD sandwiched between the two Majorana nanowires. The quantity $d_{\sigma }^{†}$$\left(d_{\sigma }\right)$ in $H_{QD}$ denotes the creation (annihilation) operator for electrons in spin state σ with a spin-dependent energy level of $\varepsilon_{\sigma }=\varepsilon_{d}+\sigma B_{z}$, in which $B_{z}$ is the vertical magnetic field which induces Zeeman splitting of the dot level. The Hamiltonian $H_{MBSs}$ denotes the two pairs of the MBSs located, respectively, in the left and right nanowires [26,27],
(1)$$\begin{array}{c} H_{MBSs}=i\sum_{\alpha =L,R}\varepsilon_{\alpha }\gamma_{\alpha 1}\gamma_{\alpha 2}, \end{array}$$
in which $\varepsilon_{L/R}$ stands for direct hybridization energy (strength) between the MBSs in the left and right nanowires. Since the MBSs are of their own anti-particles, their creation and annihilation operators are the same ($\gamma_{\alpha j}=\gamma_{\alpha j}^{†}\left(j=1,2\right)$) and obey the commutation relation of $\left\{\gamma_{\alpha i},\gamma_{\alpha^{\prime }j}\right\}=\delta_{\alpha ,\alpha^{\prime }}\delta_{i,j}$ [21,39]. Couplings between the QD and the MBSs are accounted for by $H_{T}=\sum_{\alpha =L,R}H_{d\alpha }$, with [26,27]
(2a)$$\begin{array}{ccc} & & H_{dL}=\sum_{\sigma }\left(\lambda_{L\sigma }d_{↑}-\lambda_{L\sigma }^{*}d_{↑}^{†}\right)\gamma_{L1}, \end{array}$$
(2b)$$\begin{array}{ccc} & & H_{dR}=i\left[\lambda_{R}\left(\cos\frac{\theta }{2}d_{↑}+\sin\frac{\theta }{2}d_{↓}\right)-\lambda_{R}^{*}\left(\cos\frac{\theta }{2}d_{↑}^{†}+\sin\frac{\theta }{2}d_{↓}^{†}\right)\right]\gamma_{R2}, \end{array}$$
where $\lambda_{\alpha \sigma }$ denotes the spin-dependent coupling strength between the QD and the α-Majorana nanowire [22,23,24]. Note that although only $\lambda_{L\sigma }$ depends on electron spin, we use the symbol $\lambda_{\alpha \sigma }$ for the sake of consistency. Due to the proximity of the nanowire to the superconductor substrates, a phase factor is induced and added to $\lambda_{\alpha \sigma }$ as $\lambda_{\alpha \sigma }=\left|\lambda_{\alpha \sigma }\right|\exp\left(i\phi_{L/R}/2\right)$, which is set to be $\phi_{L/R}=\pm \Delta \phi /2$ for symmetrical nanowires [26,27]. It is known that this phase difference is responsible for the generation of Josephson current [26,27,28,29,32]. The symbol θ stands for the bent angle between the left and right Majorana nanowires that are not aligned. Next, we make an unitary transformation to change the Majorana fermion representation to a conventional fermion representation for the sake of calculation convenience: [20,21,26,27] $f_{L/R}=\left(\gamma_{L/R1}+i\gamma_{L/R2}\right)/\sqrt{2}$ and $f_{L/R}^{†}=\left(\gamma_{L/R1}-i\gamma_{L/R2}\right)/\sqrt{2}$. To obtain the Green’s functions for the Josephson current based on the Dyson equation method, we rewrite the system Hamiltonian as a matrix, which is [26]:(3)$$\begin{array}{c} \tilde{H}=\left[\begin{array}{ccc} \;\;{\tilde{H}}_{d} & {\tilde{H}}_{dL} & {\tilde{H}}_{dR}\\ \;\;{\tilde{H}}_{Ld} & {\tilde{H}}_{L} & 0\\ \;\;{\tilde{H}}_{Rd} & 0 & {\tilde{H}}_{R} \end{array}\right], \end{array}$$
in which the 4×4 sub-matrix ${\tilde{H}}_{d}=\varepsilon_{\sigma }{\tilde{\sigma }}_{z}$ is for electron on the QD with spin-dependent energy level $\varepsilon_{\sigma }$. In the above expression, ${\tilde{\sigma }}_{z}=\sigma_{z}⊗1_{2\times 2}$, where $\sigma_{z}$ is the usual 2×2 Pauli matrix of the *z*-component, and the symbol ⊗ denotes the matrix direct product. Similarly, the 4×4 sub-matrix ${\tilde{H}}_{L/R}=\varepsilon_{L/R}{\tilde{\sigma }}_{z}$ is the Hamiltonian of the two Majorana nanowires with MBS–MBS direct hybridization strength $\varepsilon_{L/R}$. The couplings between the QD and Majorana wires are
(4a)$$\begin{array}{ccc} & & {\tilde{H}}_{Ld}=\frac{\sqrt{2}}{2}\left[\begin{array}{cccc} \;\;-\lambda_{L↑}^{*} & -\lambda_{L↓}^{*} & \lambda_{L↓} & \lambda_{L↑}\\ \;\;-\lambda_{L↑}^{*} & -\lambda_{L↓}^{*} & \lambda_{L↓} & \lambda_{L↑} \end{array}\right], \end{array}$$
(4b)$$\begin{array}{ccc} & & {\tilde{H}}_{Rd}=\frac{\sqrt{2}}{2}\left[\begin{array}{cccc} \;\;\lambda_{R}\cos\frac{\theta }{2} & \lambda_{R}\sin\frac{\theta }{2} & \lambda_{R}^{*}\sin\frac{\theta }{2} & \lambda_{R}^{*}\cos\frac{\theta }{2}\\ \;\;-\lambda_{R}\cos\frac{\theta }{2} & -\lambda_{R}\sin\frac{\theta }{2} & -\lambda_{R}^{*}\sin\frac{\theta }{2} & -\lambda_{R}^{*}\cos\frac{\theta }{2} \end{array}\right]. \end{array}$$

In this manuscript, we focus on the DC Josephson current through the system, which is calculated by using the nonequilibrium Green’s function technique [19,26,27,29,32,40]:(5)Jσ=eh∫dεReTr[σ˜z(∑˜σaGdσa−∑˜σrGdσr)]f(ε),
in which ${\tilde{\sum }}_{\sigma }^{r/a}=\sum_{L\sigma }^{r/a}-\sum_{R\sigma }^{r/a}$ represents the deviation of the self-energies contributed from different Majorana nanowires, with $\sum_{\alpha \sigma }^{r/a}={\tilde{H}}_{d\alpha }g_{\alpha }^{r/a}{\tilde{H}}_{\alpha d}$ [19,26,29]. The uncoupled retarded/advanced Green’s function for the two nanowires is $g_{\alpha }^{r/a}={\left[\varepsilon -H_{\alpha }+\pm i0^{+}\right]}^{-1}$. Taking into account the self-energies, we can obtain the retarded/advanced Green’s function of the QD as [26,40] $G_{d\sigma }^{r/a}={\left[\varepsilon 1_{4\times 4}-{\tilde{H}}_{d}-\left(\sum_{L\sigma }^{r/a}+\sum_{R\sigma }^{r/a}\right)+i\gamma 1_{4\times 4}\right]}^{-1}$, where γ is a suitable Dynes broadening to avoid the divergence of the retarded Green’s function [32]. In the above expression, $f\left(\varepsilon \right)=1/[1+\exp\left(\varepsilon /k_{B}T\right)]$ is the equilibrium Dirac–Fermi function, where *T* and $k_{B}$ denote the temperature and Boltzmann constant, respectively. In the following, we consider the case of zero temperature and the Josephson current is determined by the ABSs with a superconductor energy gap.

## 3. Numerical Results

In this section, we present our numerical calculation results by setting $\lambda_{L↑}=\lambda_{0}=1$ as the energy unit with fixed $\lambda_{R}=\lambda_{0}$. We change the values of $\lambda_{L↓}$ to study the behavior of the spin-dependent Josephson current, $J_{\sigma }$. In the presence of nanowires hosting MBSs, the Josephson current arises from the phase difference as was previously studied in the absence of spin-dependent QD–MBS coupling [26,27]. Figure 1b shows the Josephson current varying with both ϕ and the bent angle θ. The present structure then reduces to that presented in refs. [26,27], and only the spin-up Josephson current flows through the system $J_{↑}\neq 0$. The Josephson current contributed from the spin-down electrons is always zero, even if the right Majorana nanowire couples to electrons of both spin directions for nonzero θ. The $J_{↑}-\phi$ in Figure 1b shows the typical sinusoidal lineshape for a small value of the bent angle θ. Under this simple condition ($\lambda_{L↓}=0,\theta =0$), we can obtain the analytical expressions for the retarded Green’s function and then the current $J_{\sigma }$ as,
(6)$$\begin{array}{c} G_{d\sigma }^{r}=\frac{1}{\left(\varepsilon -\varepsilon_{-,\sigma }-2B_{\sigma }\right)\left(\varepsilon -\varepsilon_{+,\sigma }-2B_{\sigma }\right)-4B_{\sigma }^{2}\sin^{2}\frac{\phi }{2}}\left[\begin{array}{cc} \;\;\varepsilon_{-,\sigma }-2B_{\sigma } & -2iB_{\sigma }\sin\frac{\Delta \phi }{2}\\ \;\;2iB_{\sigma }\sin\frac{\phi }{2} & \;\;\varepsilon_{+,\sigma }-2B_{\sigma } \end{array}\right], \end{array}$$
where $B_{↑}=B_{↓}=\lambda_{0}^{2}\varepsilon /\left(\varepsilon^{2}-\delta_{M}^{2}+i0^{+}\right)$ and $\varepsilon_{\pm ,\sigma }=\varepsilon -\varepsilon_{\sigma }+i\gamma$. Furthermore, the sinusoidal form of the current varying with respect to ϕ is given by
(7)$$\begin{array}{c} J_{\sigma }=-\frac{e\sin\phi }{h}\int_{-\infty }^{0}\frac{4B_{\sigma }^{2}}{|\left(\varepsilon -\varepsilon_{-,\sigma }-2B_{\sigma }\right)\left(\varepsilon -\varepsilon_{+,\sigma }-2B_{\sigma }\right)-4B_{\sigma }^{2}\sin^{2}\frac{\phi }{2}{|}^{2}}d\varepsilon . \end{array}$$

Note that at small θ, a pair of peaks with opposite signs emerge in the Josephson current around ϕ=π and −π, respectively, which is quite stable with the change in dot level, interaction between the QD and MBSs, or the magnetic field applied on the dot [26]. As will be shown in the following, when the dot couples to the Majorana nanowires with spin-resolved strengths, the jump in Josephson current at ϕ=π can be destroyed, which is very different from the case in refs. [26,27]. With increasing θ, the hybridization strength between spin-up electrons and the MBSs in the right Majorana nanowire becomes weaker and the associated Josephson current is reduced accordingly. Note that as $\lambda_{L↓}=0$, the spin-down current is always zero although electrons at this spin state couple to the MBSs at the right nanowire for nonzero θ. For θ=π, i.e., the mode of $\gamma_{R1}$ is rotated close to the dot, the value of $J_{↑}$ is zero. This is because the modes of the MBSs $\gamma_{L1}$ and $\gamma_{R1}$ cannot be paired, and thus the transport processes are terminated. The above results are consistent with those in ref. [26].

If the electrons in the QD of both spin directions couple simultaneously to the Majorana nanowires, mutual interference between them occurs and exerts a significant influences on the Josephson current. We then study the influences of $\lambda_{L↓}$ on the spin-dependent Josephson current in Figure 2 under the conditions of θ=0 and π/2, respectively. For straight Majorana nanowires (θ=0), spin-down electrons are free from coupling with the right nanowire, and thus $J_{↓}\equiv 0$. We thus only show $J_{↑}$ in Figure 2a. The jump in $J_{↑}$ at ϕ=π survives for nonzero $\lambda_{L↓}$. With increasing values of $\lambda_{L↓}$, the magnitude of $J_{↑}$ is monotonously suppressed due to the fact that the MBSs generally suppress electron transport processes [21,39]. Figure 2b shows that the magnitude of $J_{↑}$ at $\lambda_{L↓}>0$ is larger than that of $\lambda_{L↓}=0$ (solid black line). However, with increasing $\lambda_{L↓}$, the value of $J_{↑}$ varies in a complex way. For $0<\lambda_{L↓}<\lambda_{0}$, the magnitude of $J_{↑}$ increases with increasing values of $\lambda_{L↓}$, and then decreases for larger $\lambda_{L↓}$ values (the pink dash-dotted line). Such a behavior of $J_{↑}$ results from competition between spin-up and spin-down electrons interacting with the MBSs.

Figure 2c shows $J_{↓}$ at different values of $\lambda_{L↓}$ for fixed θ=π/2, i.e., the two nanowires are perpendicular. Now the distances between the QD and the two modes of MBSs $\gamma_{R1}$ and $\gamma_{R2}$ are the same, and enable the emergence of perfect constructive interference. As $\lambda_{L↓}=0$, $J_{↓}$ is zero, as was pointed out earlier. With increasing $\lambda_{L↓}$, the magnitude of $J_{↓}$ increases monotonously as spin-down electrons more strongly couple to the MBSs. We note that spin-up and spin-down electrons behave differently due to the MBSs helical properties. When $\lambda_{L↓}=\lambda_{L↑}=\lambda_{0}$ and θ=π/2, i.e., electrons of opposite spin directions hybridize to the MBSs symmetrically, we find that the sinusoidal relationship between $J_{\sigma }$ and ϕ is destroyed. Moreover, the Josephson currents of both the two spin directions are significantly enhanced and become opposite in sign. This because if spin-up and spin-down electrons couple to the left Majorana nanowire with equal strength, a constructive interference effect between them is induced. Furthermore, if θ=π/2, the distances between the two ends of the right Majorana nanowire and the dot are the same and also enable the emergence of a perfect constructive interference process. As a result of the above two cases, the Josephson current is significantly enhanced. This is the central result of this manuscript. Since now Josephson currents of opposite spin directions are almost equal in magnitude but opposite in sign, a large value of the pure spin Josephson current $J_{s}=\hbar \left(J_{↑}-J_{↓}\right)/2$ emerges in the absence of a charge current $J_{e}=\left(J_{↑}+J_{↓}\right)/2$. This is quite useful in superconductor spintronics. We have checked that this abnormal enhancement in the Josephson current happens only under the conditions of θ=π/2 and $\lambda_{L↓}=\lambda_{L↑}=\lambda_{0}$, i.e., when the MBSs hybridize to electrons of opposite spin directions in a totally symmetrical way.

Some previous work showed that if the two modes of the MBSs at opposite ends of the same nanowire interact with each other ($\delta_{M}\neq 0$), the influence of the MBSs on electronic transport will be weakened [21,34]. We show in Figure 3a that the Josephson current in the cases of θ=π/2 and $\lambda_{L↓}=\lambda_{L↑}$ becomes spin-unpolarized ($J_{↑}=J_{↓}$), and is a sinusoidal function of the phase difference for nonzero $\delta_{M}$. Such a phenomenon emerges as long as direct MBS–MBS hybridization is finite. As $\delta_{M}$ increases, the Josephson current is monotonously suppressed, with the jump at ϕ=π remaining unchanged. Figure 3b,c presents individual spin-up and spin-down currents for a fixed $\delta_{M}=0.1\lambda_{0}$ and different values of θ. The spin-up current in Figure 3b first increases with increasing θ, reaching a maximum at θ=π/2 and then decreases with further increasing θ. At fixed value of ϕ=π, the spin-up current is zero as $\lambda_{R↑}=0$. The spin-down current in Figure 3c resembles essentially the spin-up one, but is zero at either θ=0 or π. Note that only when θ=π/2, are the Josephson currents of the two spin directions the same. For other bent angles, the Josephson current is still spin-polarized due to different coupling strengths between the MBSs and the two spin states on the QD.

Experimentally, preparation of MBSs usually requires large magnetic fields applied to the nanowire [1,2,3]. These magnetic fields will unavoidably leak into the QD and change the behavior of the electrons. In ref. [26], the authors have shown that the magnetic fields along the *x*-direction will monotonously suppress the magnitude of the Josephson current. Here, we consider the impacts of a vertical magnetic field ($B_{z}$) that induces Zeeman splitting in the QD. Consideration of the vertical magnetic fields also enables us to examine the possible impacts of different dot levels on the Josephson current. Figure 4 shows that the $J_{\sigma }-\phi$ sinusoidal lineshape changes for nonzero $\lambda_{L↓}$ and $B_{z}$, except in the case of θ=π/2. The spin-polarized currents develop peaks and dips around ϕ=π depending on the value of θ. Such a result is quite different from that in ref. [26], in which $\lambda_{L↓}=0$. The abnormal change in the Josephson current arises from the interferences between spin-up and spin-down electrons when they couple to MBSs with different strengths. At last, we indicate that here we have neglected the influences of the Coulomb interaction between electrons on the QD. This is because the MBSs are of their own anti-particles and zero in charge. They mainly exert significant effects on the transport properties near the zero-energy states. Moreover, in our structure with the QD sandwiched between the two Majorana nanowires fabricated from superconductor substrates, the transport processes are determined by the states within the superconducting gap, and the Coulomb interaction can usually be neglected, as in refs. [12,18,29].

## 4. Summary

In summary, we find an abnormal enhancement in the Josephson current through a QD when electrons of both spin directions couple to the MBSs with the same amplitude. Such an exotic phenomenon emerges only when the two nanowires are arranged perpendicular with respect to each other (bent angle θ=π/2) and the two modes of the MBSs in each nanowire are decoupled ($\delta_{M}=0$). Moreover, we find that the enhanced spin-up and spin-down Josephson currents are equal in amplitude but opposite in direction, resulting in a large spin current without a charge current. This is crucial in spintronic devices based on superconductor materials. We also find that in the presence of direct MBS–MBS overlap, the Josephson current becomes spin-unpolarized and its sinusoidal lineshape versus the phase difference is restored.

## Figures and Tables

**Figure 1 nanomaterials-13-01482-f001:**
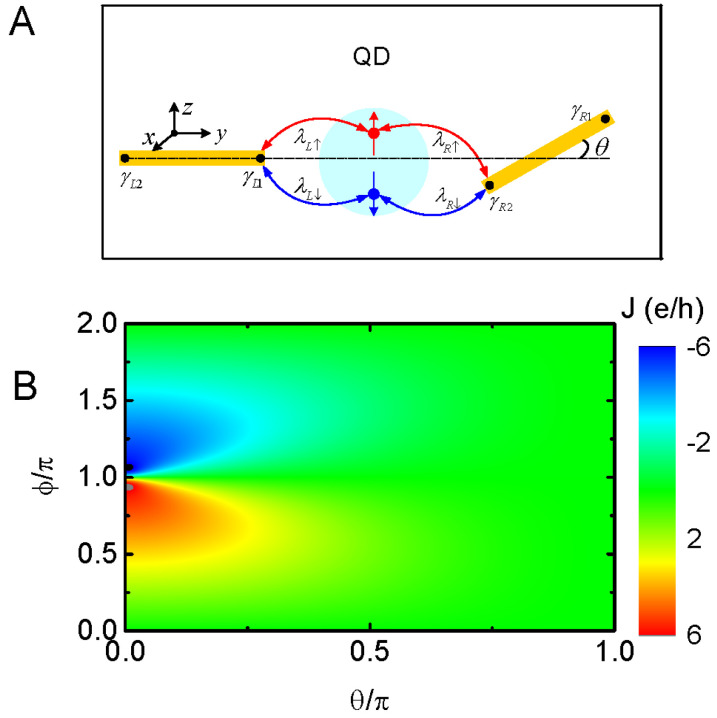
(**A**) Schematic diagram of the system with a QD (bluish circle) interacting with the MBSs prepared at the left and right Majorana wires with strengths of $\lambda_{L\sigma }$ and $\lambda_{R\sigma }$. The two nanowires form an angle of θ with respect to each other. (**B**) Contour line of the Josephson current versus the phase differences ϕ and θ with fixed $\lambda_{L↓}=\varepsilon_{d}=B_{z}=\delta_{M}=0$.

**Figure 2 nanomaterials-13-01482-f002:**
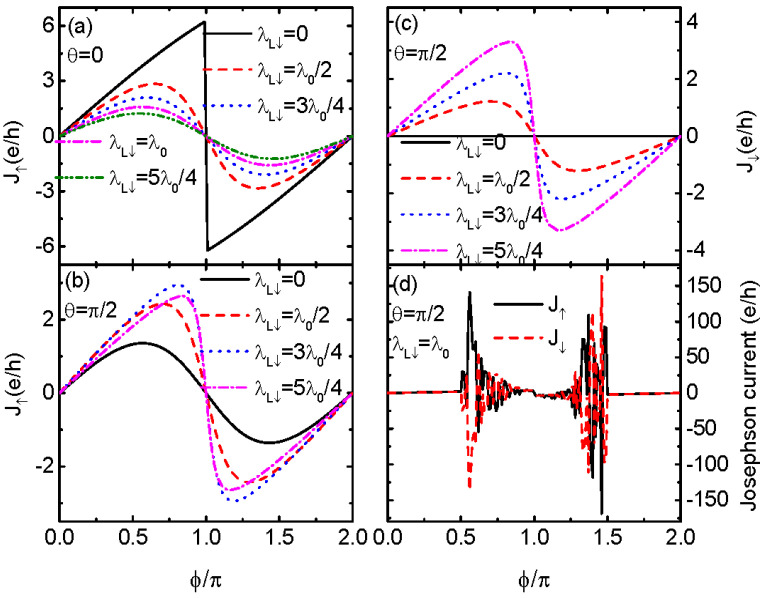
Josephson current $J_{↑}=J_{↓}$ versus ϕ for θ=0 in (**a**), $J_{↑}$ in (**b**) and $J_{↓}$ in (**c**) for θ=π/2 and different values of $\lambda_{L↓}$. Figure (**d**) shows the abnormal enhancement in the Josephson current under the conditions of θ=π/2 and $\lambda_{L↓}=\lambda_{L↑}=\lambda_{R}=\lambda_{0}$. Other parameters are as in Figure 1.

**Figure 3 nanomaterials-13-01482-f003:**
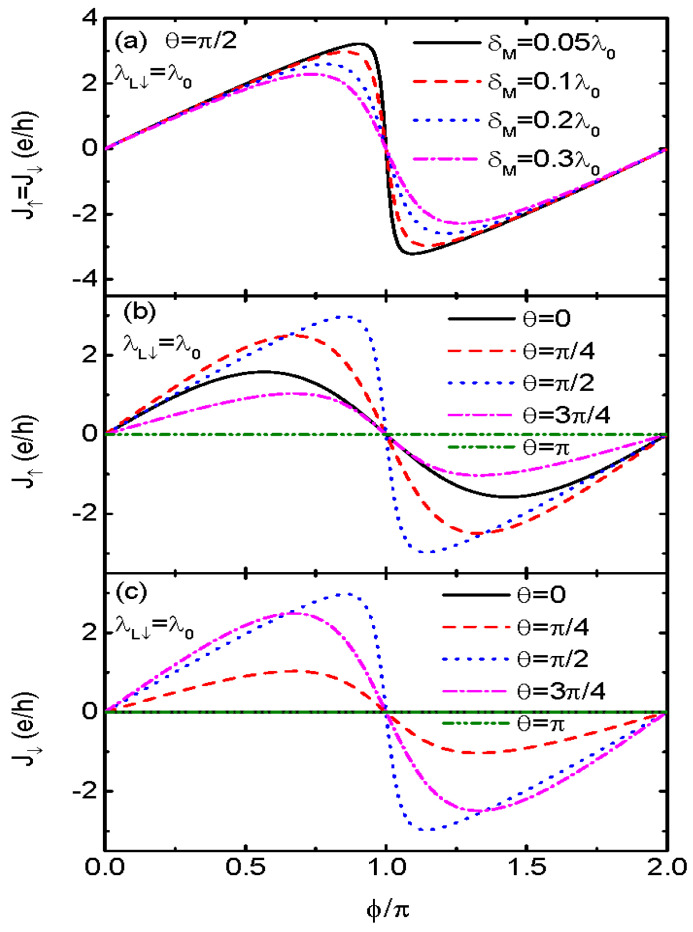
Josephson current $J_{↑}=J_{↓}$ versus ϕ for θ=π/2, $\lambda_{L↓}=\lambda_{0}$ and different $\delta_{M}$ in (**a**). Figures (**b**,**c**) are for $J_{↑}$ and $J_{↓}$, respectively, under the conditions of $\delta_{M}=0.1\lambda_{0}$ and different θ. Other parameters are as in Figure 1.

**Figure 4 nanomaterials-13-01482-f004:**
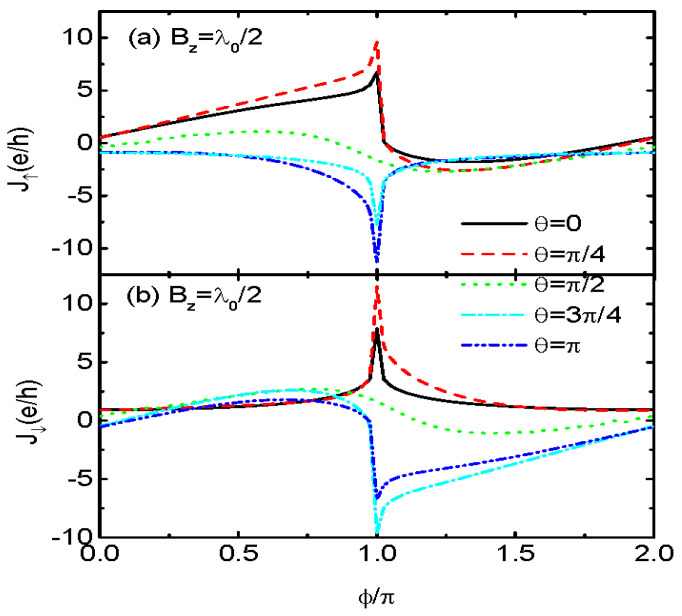
$J_{↑}$ in (**a**) and $J_{↓}$ in (**b**) for $B_{z}=\lambda_{0}/2$, $\lambda_{L↓}=\lambda_{0}$ and different values of θ. Other parameters are as in Figure 1.

## Data Availability

All data included in this study are available upon request by contact with the corresponding author.

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
