# Peer review of "Enhancement of the Josephson Current in a Quantum Dot Connected to Majorana Nanowires"

_nanomaterials, 2023, doi:10.3390/nano13091482_

Round 1

Reviewer 1 Report

The authors calculate the spin polarized currents in a Josephson junction of a quantum dot with superconductor nanowires.   They find anomalous interference or addition of these currents, depending on the coupling strength between the spin states in the dot and in the nanowires, and on the relative orientation of the nanowires described by an an angle theta.  Several aspects of this manuscript should be improved. 

1 - It is hard to understand Figure 1, especially part A.  I suggest enlarging it and changing the aspect ratio.  Both A and B subfigures look vertically compressed.

2 - It is hard to understand the element Hd (with a tilde) of the matrix Hamiltonian (3).  Better explanations are needed.

3 - How the angle theta can be controlled? 

4 - The summary (section 4) is inconsistent with Figure 2.  The angle theta is not mentioned.  What is the role of that angle?  

5 - Many text paragraphs are very long, should be split into smaller paragraphs.

6 - The English language must be carefully revised, there are some confusing sentences. For example "at the two sides of phi=pi".  

The English language must be carefully revised.

Author Response

We thank very much for the advices from the reviewer, and reply the comments in the uploadded file. Please see the attachment.

Reviewer 2 Report

 In this manuscript, the author reports, “Enhancement of the Josephson current in a quantum dot connected to Majorana nanowires”. The authors should address the following questions before getting a possible publication.

 Recommendation: Minor revisions are needed as noted.

 1.     The novelty of the present article should be discussed in the last paragraph of the Introduction section.

2.     Abstract must be enriched via valuable results which pave the way for understanding the

audiences

3.     Abbreviations should be defined at their first instance.

4.     The level of English throughout the manuscript needs language polishing. 

5.     The formatting and grammatical errors in the article need to be checked carefully.

6.     The author should write the purpose for each test in one/two sentences (in brief) before explaining the results of the characterization techniques. 

7. The authors have cited relevant references in the Introduction section; however the manuscript needs to be highlighted with recent reports further to broaden the impact

Minor editing of English language is required

Author Response

We thank very much the reviewer for his/her valuable comments, which are replied in the uploadded file. Please see the attachment.

Round 2

Reviewer 1 Report

Lines 186-187 "Fig. 2(c) shows J↓ at different values of λL↓ for fixed θ = π/2, i.e., the two Majorana nanowires are vertical with respective to each other".  Probably you mean "perpendicular to each other" (or simply "perpendicular") instead of "vertical with respective to each other".

The same at lines 248-249.

Still needs improvement.

Author Response

We thank the reviewer very much for  his/her responsible attitude on our manuscript, and the valuable advice which is totally accepted. In this revised version, we have replaced "vertical" by "perpendicular". We are sorry for the carelesses, and have improved the English with the help of a native English spoken friend.